# Spatiotemporal expression and control of haemoglobin in space

Josef Borg[1,12], Conor Loy[2,12], JangKeun Kim [2,12], Alfred Buhagiar[1],
Christopher Chin [2], Namita Damle[2], Iwijn De Vlaminck[2], Alex Felice[3],
Tammy Liu [4], Irina Matei [2], Cem Meydan [2], Masafumi Muratani [5],
Omary Mzava[6], Eliah Overbey [2], Krista A. Ryon[2], Scott M. Smith [7],
Braden T. Tierney [2], Guy Trudel [4], Sara R. Zwart [7,8], Afshin Beheshti [9,10] ✉,
Christopher E. Mason [2,11] ✉ & Joseph Borg [1] ✉

It is now widely recognised that the environment in space activates a diverse set of genes involved in regulating fundamental cellular pathways. This includes the activation of genes associated with blood homeostasis and erythropoiesis, with a particular emphasis on those involved in globin chain production. Haemoglobin biology provides an intriguing model for studying space omics, as it has been extensively explored at multiple -omic levels, spanning DNA, RNA, and protein analyses, in both experimental and clinical contexts. In this study, we examined the developmental expression of hae-moglobin over time and space using a unique suite of multi-omic datasets available on NASA GeneLab, from the NASA Twins Study, the JAXA CFE study, and the Inspiration4 mission. Our findings reveal significant variations in glo-bin gene expression corresponding to the distinct spatiotemporal character-istics of the collected samples. This study sheds light on the dynamic nature of globin gene regulation in response to the space environment and provides valuable insights into the broader implications of space omics research.

An interesting phenomenon that has been observed in studies related to space medicine and biology is the clinical phenotype associated with space anaemia. In the environment on board the International Space Station, astronauts experience a reduction in their red blood cell mass, associated with anaemia, which is however resolved after a few months back on Earth[1,2]. The mechanism behind the incidence of space anaemia is still unknown, much like another important occurrence that deals with haemoglobin gene switching[3]. The globin gene switch has long been thought to act as a paradigm for studying gene switch mechanisms for both in vitro and in vivo clinical models. Perturbations in the globin switch may lead to important clinical disorders, also known as haemoglobinopathies, that include, amongst others, sickle cell disease and beta-thalassaemia[4]. The regulation and control of the globin genes are governed by a set of transcription factors and other key molecules that, when brought together in close proximity, will dynamically shift the beta-globin locus and express the globin genes in a developmental pattern. The fine and exquisite nature of the globin gene control, as well as their extensive studies to date on all -omic

[1]Faculty of Health Sciences, University of Malta, Msida MSD2080, Malta. [2]Department of Physiology and Biophysics, Weill Cornell Medicine, New York, NY, USA. [3]Department of Surgery, Faculty of Medicine and Surgery, University of Malta, Msida MSD2080, Malta. [4]Ottawa Hospital Research Institute, Department of Medicine, Ottawa, Ontario, Canada. [5]Department of Genome Biology, Institute of Medicine, University of Tsukuba, Tsukuba, Japan. [6]Meinig School of Biomedical Engineering, Cornell University, Ithaca, NY, USA. [7]Biomedical Research and Environmental Sciences Division, Human Health and Performance Directorate, NASA Johnson Space Center, Houston, TX, USA. [8]University of Texas Medical Branch, Galveston, TX, USA. [9]Blue Marble Space Institute of Science, Space Biosciences Division, NASA Ames Research Center, Moffett Field, CA, USA. [10]Stanley Center for Psychiatric Research, Broad Institute of MIT and Harvard, Cambridge, MA, USA. [11]The WorldQuant Initiative for Quantitative Prediction, Weill Cornell Medicine, New York, NY 10065, USA. [12]These authors contributed equally: Josef Borg, Conor Loy, JangKeun Kim. ✉e-mail: afshin.beheshti@nasa.gov; chm2042@med.cornell.edu; joseph.j.borg@um.edu.mt

fronts, have made them the model of choice to study in one of the harshest environments that humans can experience.

Erythropoiesis is a dynamic multi-step process[5] involving the differentiation of early erythroid progenitors to enucleated red blood cells. Under normal circumstances, a large number of transcription factors such as GATA-1, GATA-2, LMO2, FOG, C-MYB, TAL-1, BCL11A, RUNX-1 and PU.1 and KLF1 are involved in the formation of haematopoietic cell lineages that are in turn finely controlled and regulated in producing a subset of cells capable of carrying vast quantities of the haemoglobin protein for oxygen transport to all tissues[6,7].

In this study, we used multiple omic datasets retrieved from several different human astronaut missions using the powerful platform made available by NASA's GeneLab[8]. This platform is now an established space omics database that provides a wealth of information linking molecular biology and genetics with space medical sciences. In particular, the full astronaut blood, as well as transcriptional data from the NASA Twins Study[9,10] was analysed for the globin gene expression profile as well as the extensive list of transcription factors and miRNAs involved in the globin gene switch mechanism.

Our research uncovered two important conclusions. Firstly, exposing humans to orbiter effects resulted in a significantly down-regulated expression of all globin genes, seemingly confirming space anaemia as documented earlier[1]. The levels of expression were then returned back to normal post-flight. Key genes that play important roles in erythropoiesis were also highly perturbed during spaceflight, with some undergoing up-regulation and others down-regulated, indicating that space anaemia might occur as a combination of direct haemolysis coupled with reduced overall production of new, mature erythroid cells.

Secondly, it was noted that an adult-to-fetal globin switch, similar to that observed in the amelioration of haemoglobinopathies on Earth, can be observed through the up-regulation and down-regulation of a number of genes, encoding for different globins, transcription factors and other key molecules that play a known role in globin switching mechanisms. This leads to the possible conclusion that the microgravity environment might elicit the mechanisms behind the developmental globin gene switch model and sheds an even more in-depth look at the key players involved in this intricate switch. These may constitute an important observation for those next frontier missions that look beyond the moon and further into deep space.

## Results

Obtained globin gene, transcription factor (TF) and globin miRNA expression results are hereunder showcased for the NASA Twins study, the JAXA study as well as the Inspiration4 mission. A summary of observed trends is provided in Fig. 1.

Notably, repressors for fetal haemoglobin (HBF) in the haemoglobin gene switch showed in-flight down-regulation in almost all cases for both the NASA twins study and the JAXA astronaut cohort, suggesting an in-flight shift to favour the production of HBF. Erythropoiesis promoters showed a mixed in-flight and post-flight expression regulation, with some promoters down-regulated in flight and thus likely contributing to the space anaemia phenotype while other promoters were up-regulated in flight, possibly in response to space anaemia. Erythropoiesis repressors were down-regulated in flight, possibly as a response to the space anaemia condition as well.

For the NASA Twins study, which constitutes a one-year mission on the ISS, globin gene and TF expression information is shown as comparisons carried out between the ground and flight twin (pre-flight), as well as comparisons made for the flight twin at different stages in the mission (i.e. pre-flight, during flight and post-flight). Different cellular fractions are separately considered, with the LD and the CPT (mononuclear cells separated by centrifugation) cell types representing the erythroid-containing cellular fractions. Due to the large number of genes and TFs considered, only those showcasing a

significant down-regulation or up-regulation ($p$-value < 0.1) in at least one comparison instance are included in Fig. 2. Separately indicated are genes and TFs in cellular fractions that showed a higher level of differentially expressed significance, with a $p$-value < 0.05.

Despite the primary focus being on haemoglobin control, regulation, and expression in the context of space-related anaemia, preserving and presenting data on white blood cells (WBCs) is pivotal for a comprehensive understanding of physiological changes in astronauts. WBC parameters offer valuable insights into the broader immune response, potential inflammation, and overall haematological health during space missions. Considering the interconnecting nature of erythroid and immune systems, the analysis and data of WBCs can provide crucial context to haemoglobin dynamics, ensuring a holistic interpretation of the physiological adaptations astronauts undergo in the space environment. This integrative approach enhances the scientific significance of our study and contributes to a more thorough comprehension of the impact of space travel on multiple facets of human physiology. Thus, while the erythroid-containing cellular fractions are perhaps most directly pertinent, the WBC fractions are still of relevance to this study.

For the JAXA cohort, which constitutes of data from 6 astronauts who spent a 6-month period on the ISS, normalised expression values are provided for all globin genes and TFs, including timestamps for temporal resolution of specific expression readings collected in-flight and post-flight. In Fig. 3, up-regulation or down-regulation of expression for specific genes or TFs of interest can be deduced when compared to the normalised pre-flight values, also provided. Values showcased in these diagrams represent mean expression levels measured across the JAXA cohort, with expression values obtained from cell-free RNA-Seq samples.

Following differential expression analysis using DESEQ2, volcano plots of $p$-values against log2 fold-changes were computed for the JAXA study, comparing pre-flight, flight and post-flight expression readings as shown in Fig. 4. Genes are known to be involved in erythropoiesis and the globin gene switch mechanism, were labelled accordingly if found to be significantly up-regulated or down-regulated. Obtained differential expression values show that erythropoietin (EPO) was indeed significantly down-regulated in-flight, when compared to pre-flight levels, but was then up-regulated again post-flight, when compared to in-flight levels.

Globin genes such as HBA1, HBB and HBD are also shown to be significantly up-regulated post-flight, when compared to in-flight levels, with levels of post-flight HBA1 being also significantly up-regulated when compared to pre-flight levels. The same observation was made for ALAS2, which seems to have been significantly up-regulated post-flight when compared to both pre-flight and in-flight expression levels. Conversely, HBE1 was significantly up-regulated in-flight, when compared to pre-flight levels. Notably, KLF1 was found to be significantly up-regulated post-flight, when compared to in-flight expression levels, which is of notable significance for the globin gene switch mechanism, as further discussed in discussion section.

Principal component analysis (PCA) was carried out accordingly for the JAXA astronauts. Specifically, PCA was performed across different features (normalised gene expressions), grouping astronaut sample data into before, during and after flight groups. The features considered are genes of interest, that is genes known to be involved in erythropoiesis, haematopoiesis and the globin gene switch. However, since cell-free RNA data was used for the JAXA cohort study, trans-acting factors and globin gene expressions are separately considered features of interest. Indeed, it is expected that looking at trans-acting factors would showcase a better representation of true variation between samples since globin gene expression would be expected to be representative when considering cellular expression (particularly in the red blood cell component, where globin gene expression is most important).

There is an observable separation between the three groups when considering trans-acting factors, with considerable variance observed

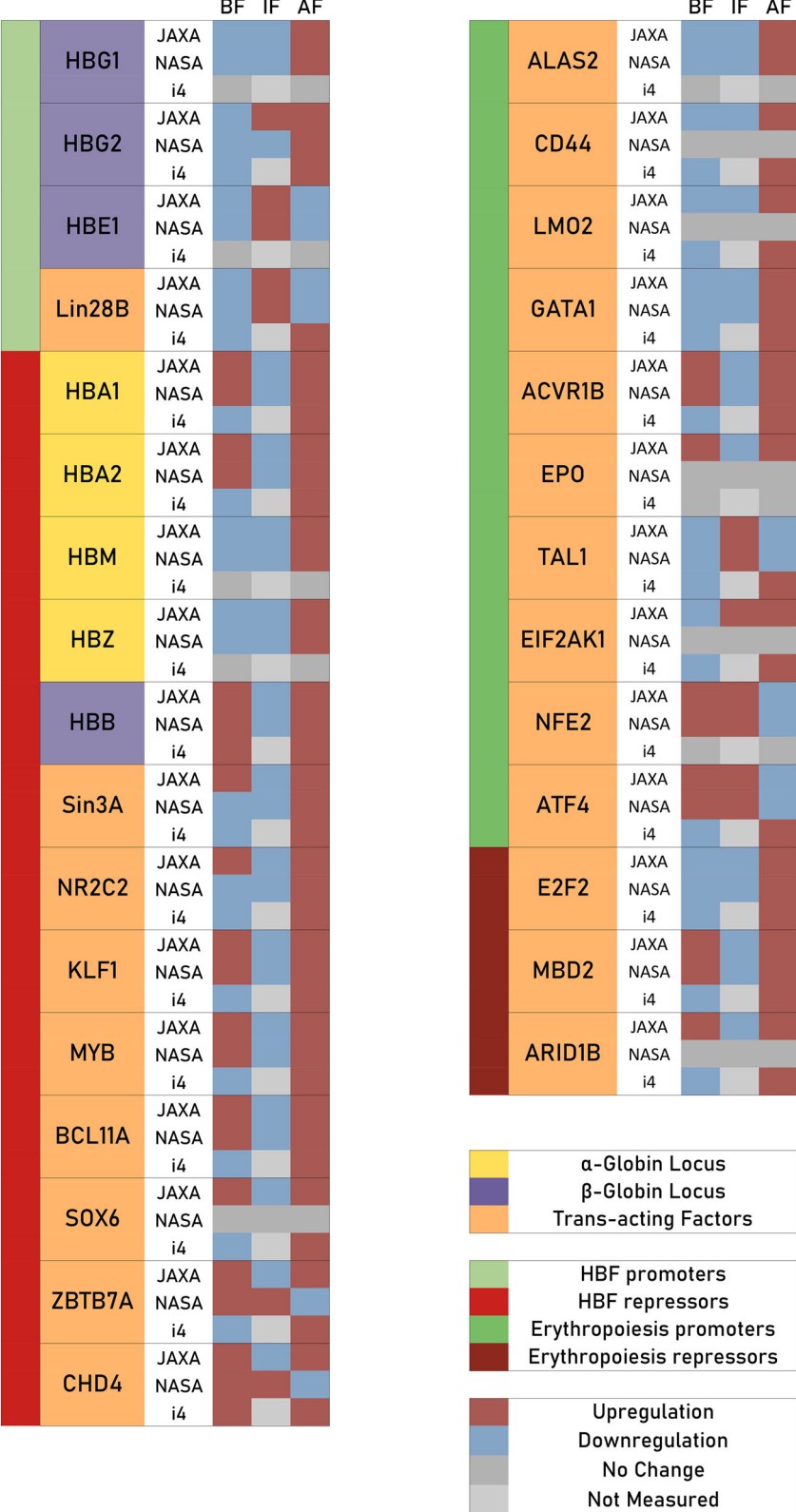

**Fig. 1 | Summary of observed differential expression in genes implicated in the globin gene switch mechanism and erythropoiesis.** A summary figure showcasing observed expression in a number of genes and transcription factors of interest, with observed trends across all three different astronaut cohorts for samples obtained before flight (BF), in flight (IF) and after flight (AF). A number of genes relevant to the globin switch mechanism are shown on the left, while a number of selected genes of relevance to erythropoiesis are shown on the right.

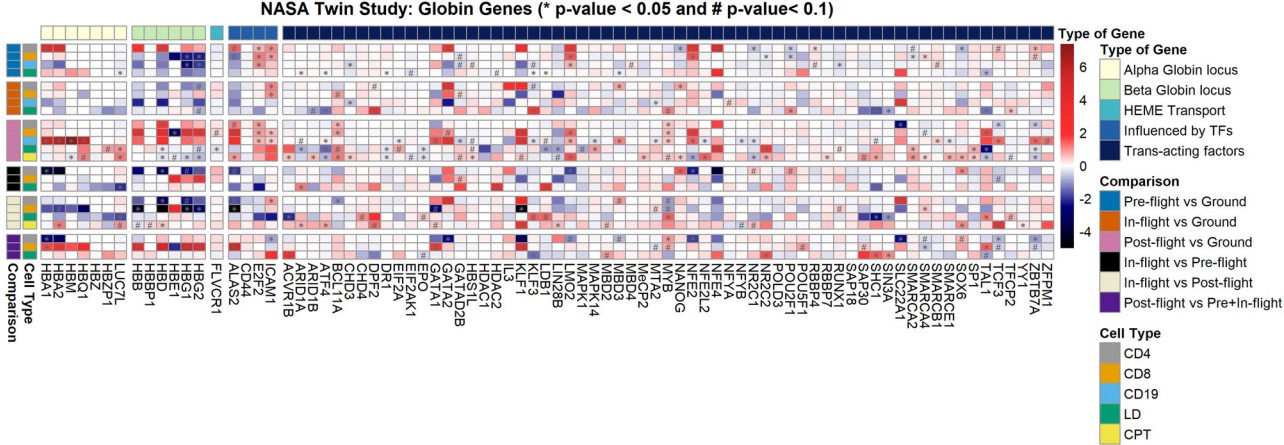

**Fig. 2 | Diagram showcasing comparisons in expression, for a number of different cellular fractions, in the NASA Twins study.** Comparisons were made both between the ground and flight twin as well as comparisons made on a spatio-temporal basis for the flight twin at different mission stages. Only genes showcasing at least one expression comparison with significant variation (*p*-value < 0.1) are included in the diagram, to maintain diagram clarity. Genes with *p*-values < 0.05 are separately indicated. All log2 fold-changes and *p*-values were calculated using DESeq2 software. Source data for the figure is provided in the Source Data file.

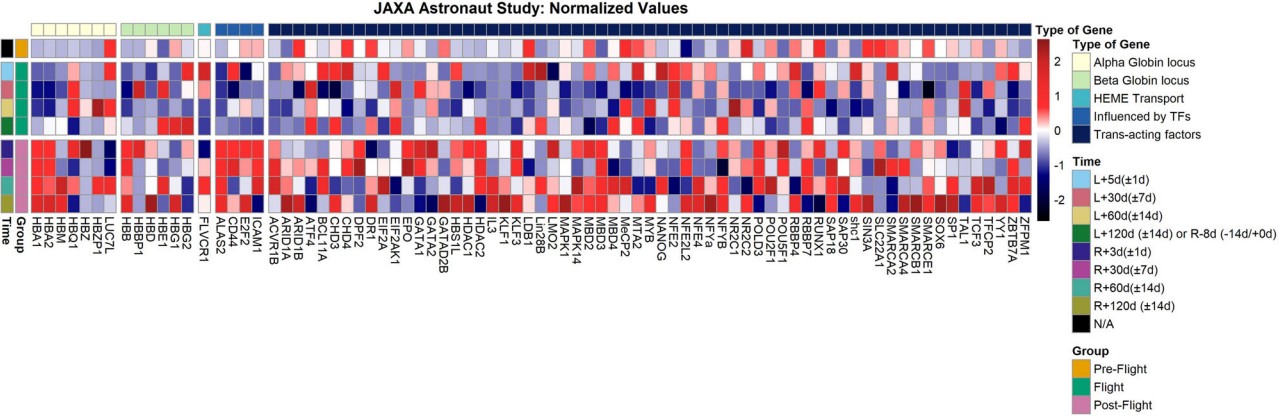

**Fig. 3 | Normalised expression values for known globin genes and other trans-acting factors for the JAXA astronaut cohort.** Genes and trans-acting factors considered were those known to be involved in the globin gene switch mechanism and/or erythropoiesis, and expression values were obtained before, during and after spaceflight, at specific intervals, for the JAXA astronaut cohort, which was composed of six JAXA astronauts. Source data for the figure is provided in the Source Data file.

across multiple principal components in this regard. This is particularly evident when looking at PC1 and PC2, which collectively account for approximately 68.3% of the variance measured across the data. There is also segregation of samples by flight status when looking at globin genes, particularly along the first principal component, which accounts for almost 99% of all variance measured in the globin gene data. These plots are provided in Fig. 5.

A diagram of normalised expression levels is provided for the Inspiration4 crew in Fig. 6. For this particular case, expression values are only available pre-flight and post-flight, with no in-flight measurement data available. The cellular type here assessed constitutes the peripheral blood mononuclear cells (PBMCs) cellular fraction. As with the JAXA cohort data, up-regulation and down-regulation post-flight can be inferred from direct comparisons with normalised expression levels pre-flight. Post-flight readings are temporally resolved.

A heat-map of *z*-scores for normalised cell-free RNA abundance in pre-flight and post-flight samples for the Inspiration4 crew is shown in Fig. 7, with alpha globin genes, beta-globin genes and trans-acting factors separately shown. Since this analysis was carried out using data from cell-free plasma samples, trans-acting factors are also to be considered as more representative of spatiotemporal effects. Indeed,

the heat-map showcases an observable difference in expression between pre-flight and post-flight time points, with overall up-regulation in most trans-acting factors.

In addition, a PCA investigation on results obtained from cell-free RNA samples for the Inspiration4 crew was carried out, with some separation in samples visible by flight status as observed in Fig. 8 when taking into account genes of interest (globin genes and trans-acting factors). This indicates that, as expected, there is an observed difference in expression when comparing pre-flight against post-flight samples using only genes of interest, indicating that there is a significant effect on the expression of globin genes and trans-acting factors involved in erythropoiesis, haematopoiesis and the globin gene switch mechanism. Specific examples concerning the observed shift are provided in the discussion section.

Globin miRNA expression data was also collected for the NASA Twins study. As with the globin gene and TF expression, several cellular fractions were separately investigated and cross-compared between the ground twin and the flight twin at different mission stages. These are showcased in Fig. 9, but the ground twin data, in this case, was only included for reference, with globin miRNA expression varying notably

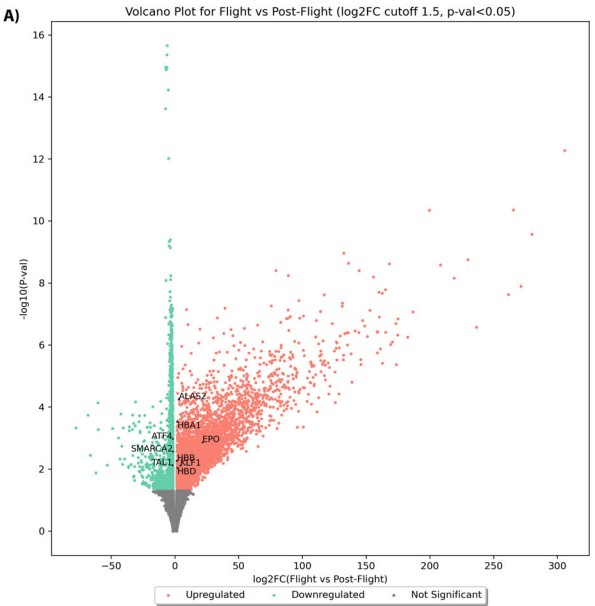

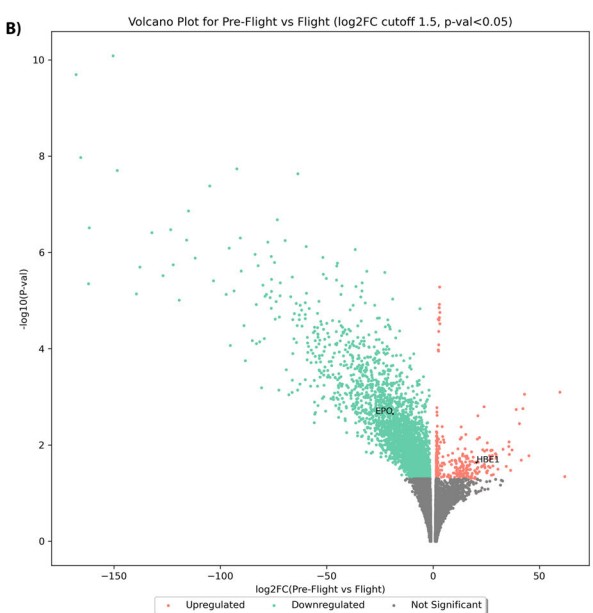

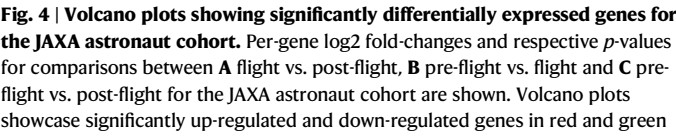

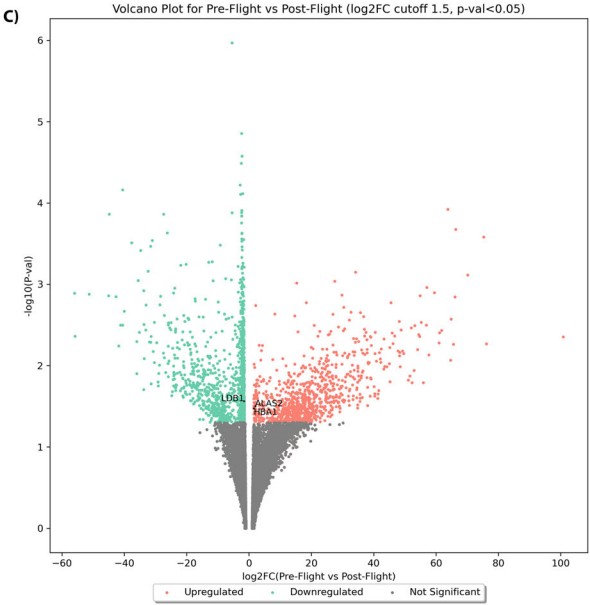

**Fig. 4 | Volcano plots showing significantly differentially expressed genes for the JAXA astronaut cohort.** Per-gene log2 fold-changes and respective *p*-values for comparisons between **A** flight vs. post-flight, **B** pre-flight vs. flight and **C** pre-flight vs. post-flight for the JAXA astronaut cohort are shown. Volcano plots showcase significantly up-regulated and down-regulated genes in red and green respectively, with genes of interest falling within the significantly differentially expressed gene sets indicated in the figure by gene symbol. All log2 fold-changes and *p*-values were calculated using DESeq2 software. Source data for the figure is provided in the Source Data file.

between the ground twin and the flight twin even before the flight, meaning that individual-specific variations were not negligible.

Physiological data collected from several different astronauts across different missions is also provided in Fig. 10). An indication of the number of astronauts that contributed to the mean data points obtained is also included.

Although not included in Fig. 10), reference can also be made to other physiological parameters that have been measured in other studies and that are of relevance to considerations made in this study. Specifically, levels of reticulocytes and erythropoietin have been measured as shown in[11], amongst others. In particular, measured red blood cell counts showcase the advent of space anaemia, with post-flight levels measured immediately upon return (and even 30 days

after return) being lower than pre-flight levels (with some recovery apparent after 30 days). In congruence with this observation, a very similar trend in levels of measured haemoglobin is also showcased.

In summary, for the JAXA astronauts, normalised mean values of expression from cell-free RNA samples for 6 different astronauts were obtained for different genes, specifically looking at gene clusters pertaining to loci of interest in the context of haemoglobin switching, as shown in Fig. 6. Volcano plots indicated particular, significantly differentially expressed genes of interest for the JAXA cohort (Fig. 4), with PCA analysis showcasing that separation by flight status is noticeable when looking at specific genes and trans-acting factors of interest in erythropoiesis, haematopoeisis and the globin gene switching mechanism (Fig. 5).

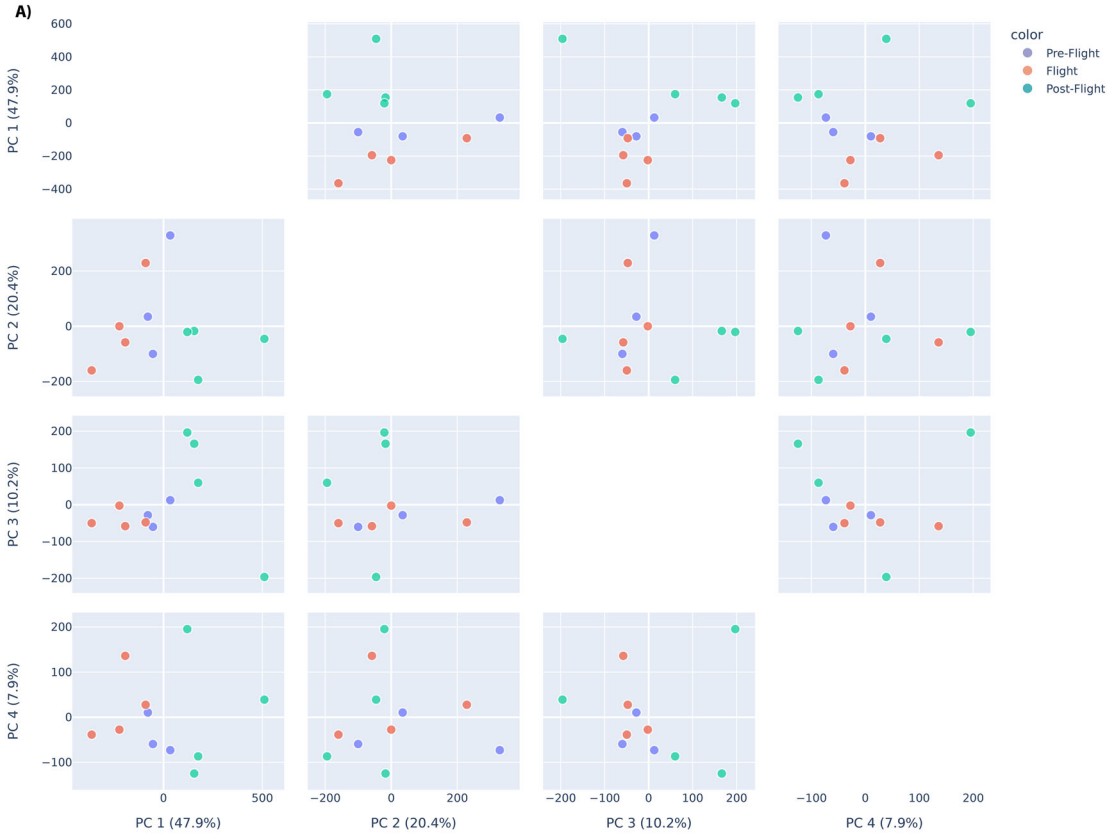

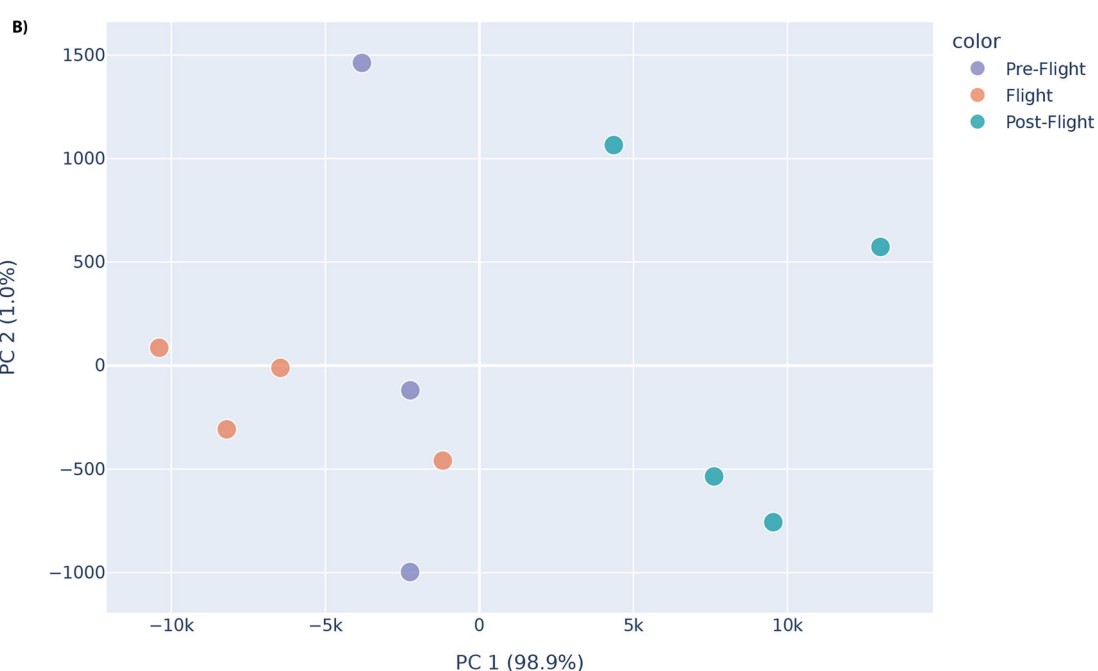

**Fig. 5 | PCA plots showing separation between pre-flight, in-flight and post-flight JAXA astronaut samples, based on variation in expression for genes of interest.** Separation between pre-flight, in-flight and post-flight samples is primarily visible along PC1, using **A** trans-acting factors and **B** globin genes as features for the JAXA astronaut cohort. For the trans-acting factors, four principal components are shown due to variance percentage contributions. Separation by flight status is visible in the PC1 vs PC2 plots for both globin genes (bottom) and trans-acting factors (top). Source data for the figure is provided in the Source Data file.

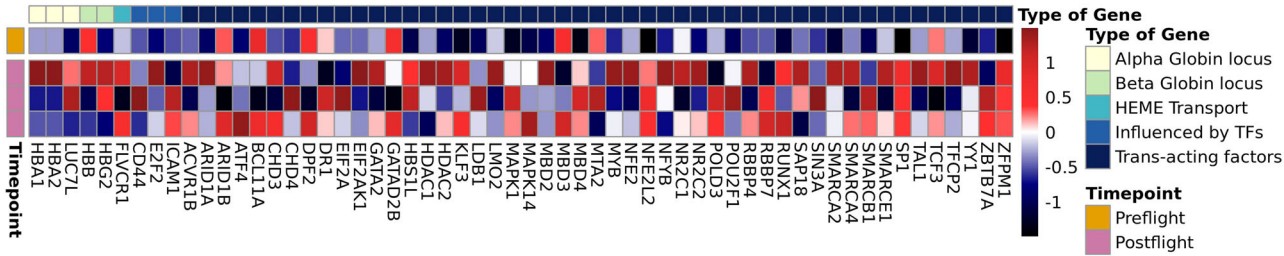

**Fig. 6 | Normalised expression values for select globin genes and trans-acting factors for the Inspiration 4 crew.** Genes and trans-acting factors selected are those known to be involved in the globin gene switch mechanism and/or erythropoiesis before, before and after spaceflight, at specific intervals, for the PBMCs cellular fraction for the Inspiration4 astronaut crew. Source data for the figure is provided in the Source Data file.

In both the NASA twins study (Fig. 2) and for the Inspiration4 crew (Fig. 6), specific blood cellular fractions were separately analysed for expression of different genes, with the LD and CPT fractions being considered as most relevant for investigation of alpha and beta-globin gene expression in the NASA Twins study, given that they are expected to contain the erythroid blood component. For the NASA twins study, differential expression analysis showcased particular genes of interest with significant up-regulation and down-regulation between different flight status samples (Fig. 2). However, relevant observations are also quoted from the non-erythroid cellular fractions, especially where trends were congruent with those observed in cell-free RNA-seq results for the JAXA astronauts. Such comparisons are expected to be of more importance when looking at transcription factors and are done while keeping in mind other physiological data as obtained across several different astronaut cohorts, as provided in Fig. 10.

## Discussion

The switch from fetal to adult haemoglobin after birth is well-known but not fully understood. Certain diseases show a reversed gene switch, with higher fetal haemoglobin linked to milder symptoms. In microgravity, astronauts experience a decrease in red blood cells, possibly due to bone mineral density loss[12,13]. This space-induced mild anaemia is intriguing for understanding mechanisms seen in clinical anaemia. The research looked into genes involved in haemoglobin production, noting similarities between astronaut and clinical anaemia in blood cell count changes.

### Gene expression analysis for the alpha and beta-globin loci

At the alpha globin coding locus, a strong trend of down-regulation during flight was noted for the JAXA astronauts for the HBA1 and HBA2 genes (for the duration of readings from 5 days after launch to 60 days after launch), followed by an observed up-regulation in expression immediately upon return to ground, even higher than expression recorded pre-flight. A similar up-regulation was observed in the NASA Twins study, where an increase in expression of the two genes was noted in the CD19 cellular fraction when comparing pre-flight and post-flight conditions. For HBA1, this was complemented with a significant strong down-regulation observed in CD4 cells when comparing in-flight against pre-flight conditions. In the CD8 fraction, post-flight readings showed up-regulation even when comparing it with combined pre-flight and in-flight values. For HBA1 and HBA2 expression in the Inspiration4 cohort, significant up-regulation from pre-flight levels was reported post-flight immediately upon return, with levels returning to pre-flight levels in subsequent post-flight readings.

These observations seem to suggest that spaceflight down-regulates expression of genes involved in alpha globin protein production, with post-flight up-regulation suggesting a recovery period (which period seemed to last until even 120 days after the astronauts' return). This up-regulation upon return also plays a role in the response to an overall loss of red blood cell count, a well-documented occurrence in astronauts (referred to as space anaemia) which manifests itself as a mild form of anaemia, as showcased by a slight lowering of red blood cell count across several different astronauts in Fig. fig:expression_physiology. It is to be noted that mutations in the HBA1 and HBA2 gene in patients is normally manifested as alpha-thalassaemia, an inherited disorder that can range in severity from a mild form of anaemia to very severe anaemia in combination with several other clinical characteristics, such as hepatosplenomegaly and cardiovascular deformities, typically leading to fetal death[14,15].

An observation of note was made in the JAXA study with HBM and its paralog, HBZ, which is involved in the production of the zeta-globin polypeptide in embryos[16,17]. While expression for both remained low in flight, expression of HBZ increased briefly immediately upon return while expression of HBM had a delayed up-regulation, with increased expression only recorded 60 days after return to ground. Strong HBM up-regulation was also recorded post-flight for the flight twin when comparing the CD19 cell fraction with the ground twin.

On the beta-globin coding locus, for the JAXA cohort, the HBB gene shows a reduction in expression observed during flight, followed by an increase in expression post-flight, surpassing expression even beyond what was observed pre-flight and mirroring observations made for HBA1 and HBA2 in the alpha globin coding locus. The HBB gene is known to directly code for the production of beta-globins, an essential sub-unit in adult haemoglobin. This seems to suggest that post-flight recovery for astronauts in this study seemed to involve enhanced expression of HBB, possibly in an attempt to recover previous repression experienced through spaceflight or as part of an overall recovery of red blood cell count in RBC-depleted astronaut blood. For the NASA Twins study, similar up-regulation was seen for HBB when comparing post-flight CD8 cell fractions with ground twin levels, with similar observations for HBG1 and HBG2. For the Inspiration4 crew, HBB gene expression was higher than pre-flight levels immediately after return but then returned to pre-flight levels.

Additionally, in-flight readings for HBG1 and HBG2 in the NASA Twins study showed down-regulation when compared to pre-flight and post-flight levels, for both the CD4 and CD8 cell fractions, as was also the case for HBB expression. Interestingly, such observations were not made for HBG1 in the JAXA CFE study, where more sporadic regulation fluctuations were observed. It was however observed, for the JAXA astronauts, that an up-regulation of HBG2 was recorded in flight. HBG2 is normally expressed in fetal bone marrow, liver and spleen, while HBE1, the epsilon globin gene, is normally expressed only at the embryonic stage. For HBE1, a similar up-regulation in flight was observed in both the JAXA astronaut cohort and in the NASA Twins study, specifically when considering the CD8 cell fraction. HBG2 was also up-regulated in post-flight readings for the Inspiration4 crew when compared to measured pre-flight levels.

Such observations seem to suggest that a globin switch is also activated in the beta-globin coding locus, with a preference for globin chains with higher oxygen affinity than those observed in embryonic and fetal haemoglobin. It is hence reasonable to suggest that similar

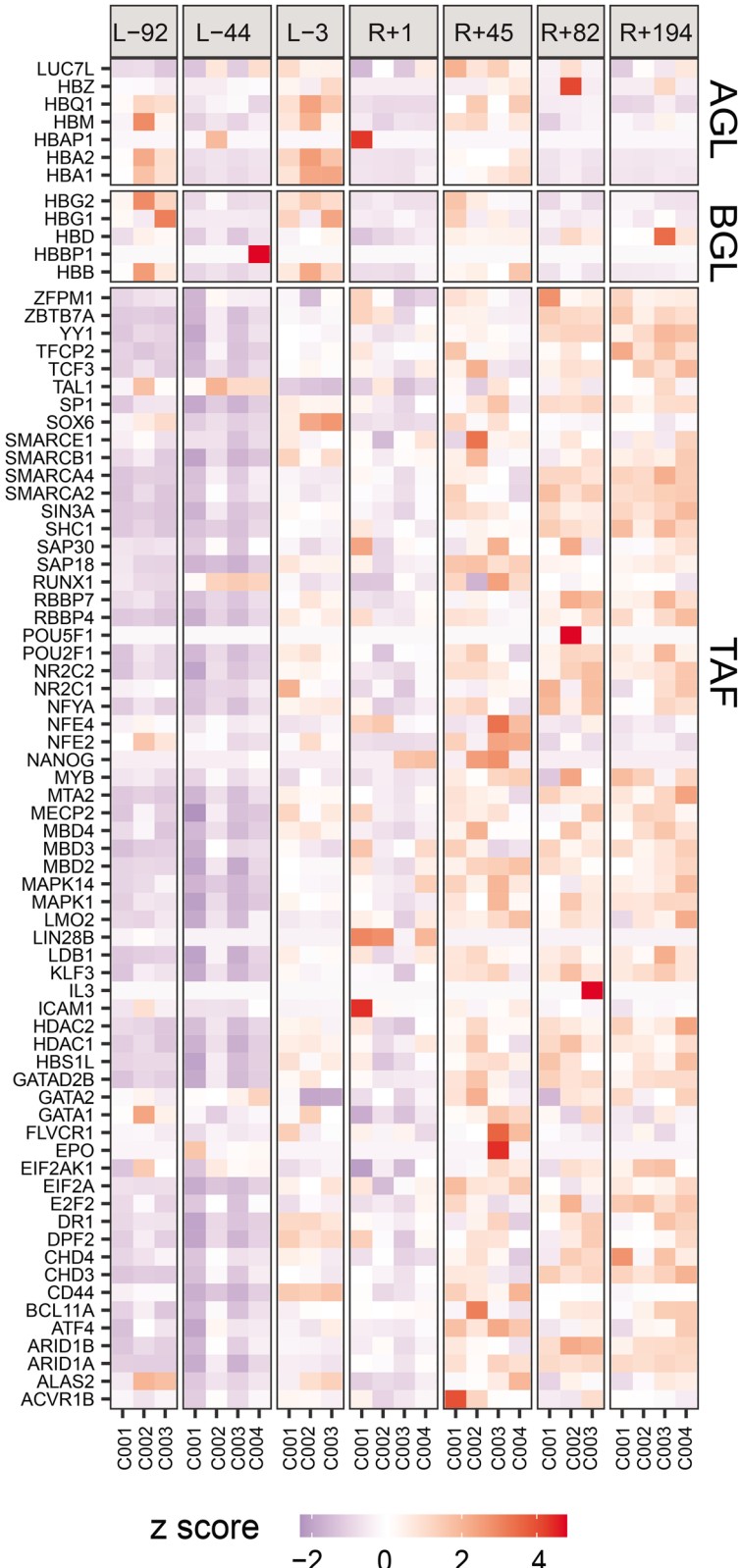

**Fig. 7 | Heat-map showcasing *z*-scores for normalised expression of several different globin genes and trans-acting factors of interest for plasma cell-free samples, with pre-flight/post-flight status indicated, for the Inspiration4 crew.** L− indicates days before flight, and R+ indicates days after flight for different obtained samples from the crew. Genes and trans-acting factors showcased are those known to be involved in the globin gene switch mechanism and/or ery-thropoiesis. Source data for the figure is provided in the Source Data file.

observed changes in other genes or transcription factors may also have an effect on the regulation of such a globin switch.

## Gene expression analysis for other erythropoietic/globin-switch genes and associated transcription factors

A cohort of other genes, known to be involved in a number of different pathways in erythropoiesis, was investigated. The ALAS2 gene was found to be heavily up-regulated immediately after the JAXA astronauts' return, an observation also mirrored in the NASA Twins

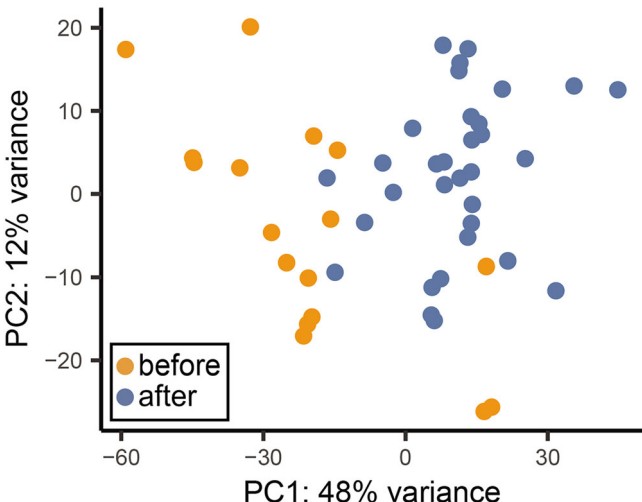

**Fig. 8 | PCA plot showcasing separation between pre-flight and post-flight samples for the plasma cell-free samples from the Inspiration4 crew.** The first two principal components are shown, accounting for around 60% of all variance across the features of interest in the data (globin genes and trans-acting factors) for the plasma cell-free samples obtained from the Inspiration4 crew. A separation between pre-flight and post-flight samples along the first principal component is discernible. Source data for the figure is provided in the Source Data file.

study, where increased expression was observed post-flight when comparing to ground twin levels in CD4, CD8 and CPT cellular fractions, together with up-regulation when comparing to the flight twin's expression before and in-flight combined in CD8 cells. The gene is involved in the development stage for erythroblasts, specifically in the production of 5'-aminolevulinate synthase 2[18] - it is indeed involved in the catalysis of this haem precursor[19]. Its up-regulation might indicate an increased rate of erythrocyte production in post-flight recovery. A similar increase post-flight was noted in the expression of the CD44 gene, known to be involved in erythropoiesis. Previous studies with patients suffering from myelodysplastic syndromes, wherein red blood cells do not achieve full maturity in the bone marrow, have shown an increased expression of CD44 in sick individuals[20]. Down-regulation in-flight and up-regulation post-flight were also observed in both the NASA Twins study (when looking at the LD cellular fraction) and the JAXA cohort when considering the E2F2 gene, which is a known negative regulator of haem oxygenase-1 (HO-1) - with enhanced expression of E2F2 resulting in lowered HO-1 levels and vice versa[20]. Since HO-1 cat-alysers the degradation of haem, the reduction of expression of E2F2 in-flight may be one of the contributing factors resulting in space anaemia. For the Inspiration4 crew, post-flight up-regulation was also observed for E2F2 in both PBMCs and cell-free RNA.

FLVCR1 was shown to have perturbed expression with phases of up-regulation observed both in-flight and post-flight in the JAXA cohort and also in post-flight readings taken for the Inspiration4 crew. This gene is known to be crucial, particularly in early erythropoiesis and encodes for two haem exporters, with FLCVR1a being involved in the expansion of erythroid cell progenitors and FLCVR1b being crucial for differentiation of said progenitors into mature erythroid cells[21]. Indeed, reduced expression of this gene is particularly observed in Diamond Blackfan anaemia[22]. Its up-regulation in both flight and post-flight observations could be seen as a response to ameliorate space anaemia, although such up-regulation was not observed in the NASA Twins study.

Transcription, or trans-acting factors, refers to elements that act to control and regulate gene expression through binding on specific

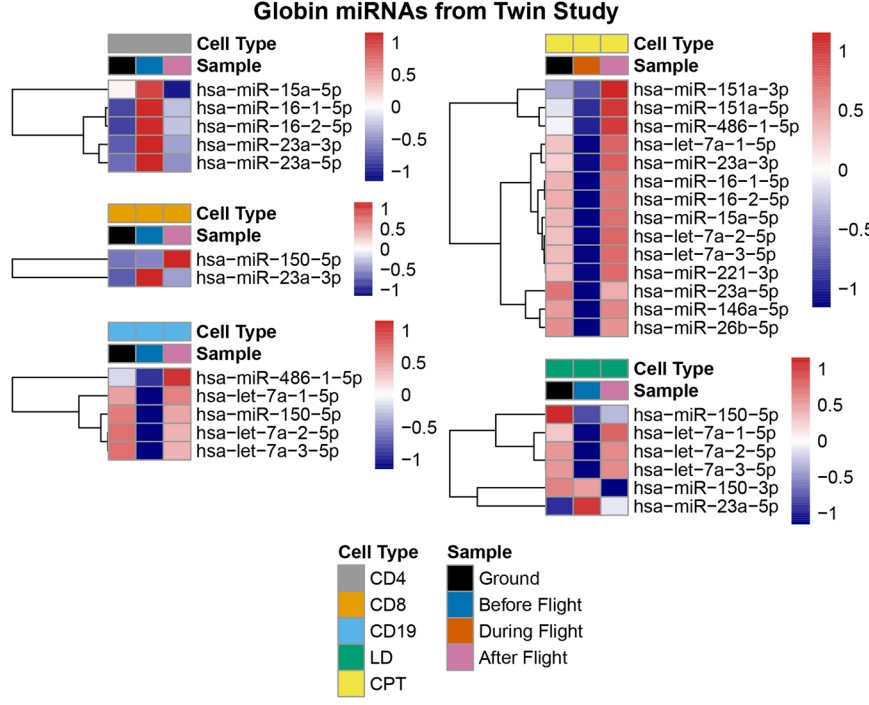

**Fig. 9 | Diagram showcasing normalised values of globin miRNA expression in the NASA Twins study.** Selected spatiotemporal comparisons of a number of globin miRNAs in different cellular fractions are presented. Source data for the figure is provided in the Source Data file.

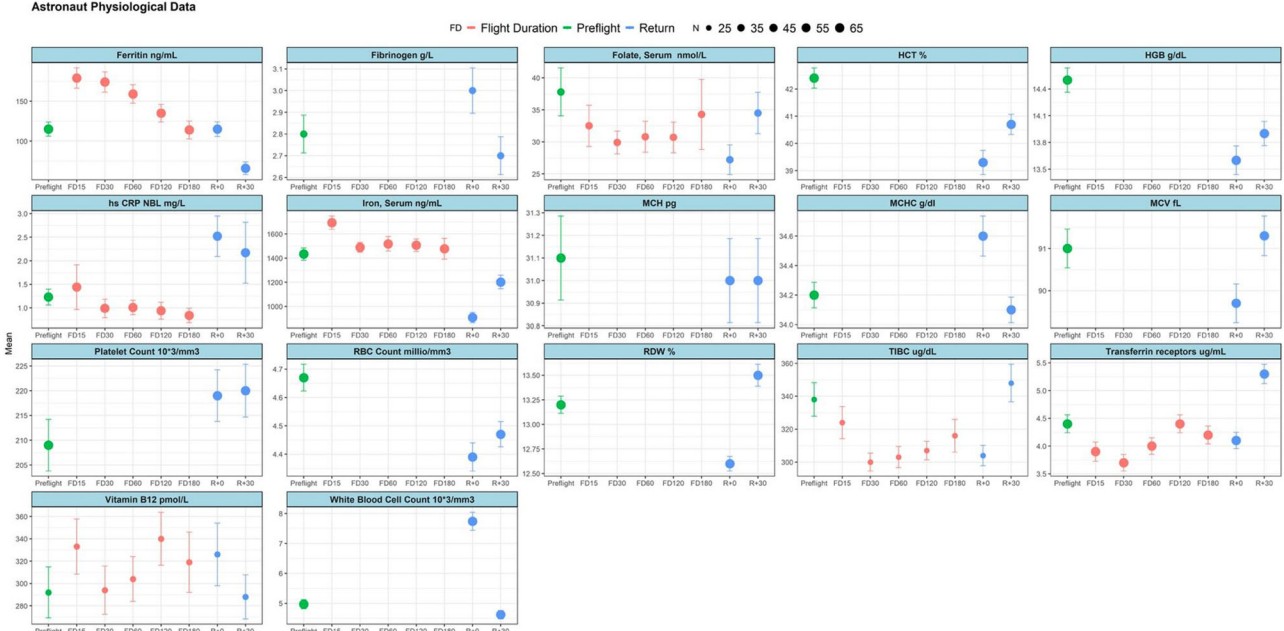

**Fig. 10 | Diagram showcasing a number of physiological parameters for a cohort of astronauts from across multiple missions.** The figure indicates levels of different parameters before, during and after flight and also denotes the sample size for every reading, as showcased by the included legend. The number of astronauts, *n*, refers to the number of astronauts for whom a particular data-point was calculated, with a standard error (SEM) indicated for every data-point accordingly. Source data for the figure is provided in the Source Data file.

regions of the genome, referred to as cis-acting or trans-acting elements and resulting in the promotion or repression of other genes. Several such transcription factors may work in tandem, and a number of genes can have several transcription factors affecting their expression, directly or indirectly. A number of such trans-acting factors have been investigated in this study, with some having a more pronounced and direct effect in erythropoiesis. While both MBD2 and MBD3 were found to be down-regulated during normal erythropoiesis in murine model bone marrow, MBD2 down-regulation was particularly shown to be crucial for erythropoiesis[20]. MBD2 promotes the degradation of CP2c family proteins, and thus its up-regulation would result in the overall down-regulation of globin gene expression[23]. MBD2 levels were shown to be down-regulated during flight for the JAXA astronauts, with levels returning to pre-flight levels after returning to the ground. This suggests a reaction to the increased rate of haemolysis known to be experienced in flight. However, it must be noted that overall globin gene expression was noted to decrease, rather than increase, during flight. A number of other trans-acting factors showed an increased expression in the first few days after arriving at the ISS, with subsequent in-flight levels going down. It has been shown, in separate studies, that a significant reduction in red blood cell count is observed in the first few days of spaceflight, after which red blood cell count stabilises to almost pre-flight levels[11]. This seems to indicate that a number of trans-acting factors closely follow haemolytic and red blood cell counts in astronaut cohorts observed in this study.

BCL11A is a well-known repressor of gamma-globin gene expression[6,24–26], with down-regulation in its expression observed in-flight in the NASA Twins study (when looking at the CD4 cellular fraction) and up-regulation noted post-flight in the same study. Down-regulation in BCL11A was also noted in flight for the JAXA astronauts. Similar observations were made for SOX6 in the JAXA astronaut cohort, which is known to coordinate gamma-globin expression with BCL11A via proximal promoter binding on the gamma-globin genes[27]. These observations suggest that BCL11A and SOX6 expression was reduced in flight, due to space anaemia, to allow the increased expression of gamma-globin, which was indeed observed to some

extent in the JAXA study but not observed in the NASA Twins study. For BCL11A, although in-flight readings were not available for the Inspiration4 astronauts, up-regulation in expression was noted post-flight. Similar observations were made in the JAXA cohort with ZBTB7A, another known down-regulator for gamma-globin gene expression[28,29], although no such observation was recorded in the NASA Twins study, where a slight post-flight down-regulation, when compared to pre-flight and in-flight levels, was noted. A post-flight increase in expression for ZBTB7A was however also observed in the Inspiration4 crew. Conversely, the expression of Lin28b was found to be somewhat increased in select measurements taken in-flight for the JAXA astronauts, as well as in the NASA twins study when looking at the CPT cellular fraction. This gene is known to be involved in the increased expression of fetal haemoglobin, with its increased expression in adult erythroblasts showing a subsequent increased expression of gamma-globin[30]. This also suggests that an overall switch towards the production of fetal haemoglobin was activated in flight.

Conversely, it was found that expression of Sin3A was down-regulated in flight in the JAXA astronaut cohort as well as in the NASA twins study (LD cellular fraction), with the gene known to be down-regulated to allow for the prevalence of higher fetal haemoglobin levels[31]. Indeed, Sin3A is known to worsen Beta-thalassaemia severity, possibly inhibiting the action of KLF10 which is known to ameliorate symptoms in a range of Beta-Haemoglobinopathies[32]. NR2C2 expression levels were also observed to decrease in flight in the JAXA cohort, which further seems to point towards an overall shift towards the expression of fetal haemoglobin. Indeed, NR2C2 is known to be involved as a direct transcriptional repressor of both embryonic and fetal haemoglobin[33–35]. The expression of NR2C2 was observed to decrease in-flight when considering the LD fraction of cells for the NASA Twins study as well. Another interesting observation was made with the CHD4 gene, with down-regulation noted during flight for the JAXA cohort, following up-regulation to pre-flight levels after return to Earth. CHD4 has been shown to increase the expression of both KLF1 and BCL11A, with its knockdown shown to result in higher expression of gamma-globin[36]. This follows the other observations made in the

same astronaut cohort concerning BCL11A and KLF1. It must be noted, however, that the same observation was not made in the NASA Twins study, with up-regulation of CHD4 actually being noted in-flight in this case. For the Inspiration4 crew, delayed up-regulation of expression of CHD4 was noted post-flight, when compared to pre-flight levels, with expression levels eventually falling back to pre-flight levels.

In addition, KLF1 showed a significant up-regulation in expression post-flight for the JAXA cohort, after having shown slight down-regulation in expression in-flight. In the NASA Twins study it was shown that a significant down-regulation of KLF1 was observed in-flight followed by up-regulation post-flight, when looking at the CD8 cellular fraction for the flight twin. The KLF1 gene encodes for Kruppel-like factor 1, a transcription factor known to be involved in positive regulation of the BCL11A gene[6,25]. It has also been shown that mutations in the beta-globin CACC box prevent KLF1 binding, causing $\beta$-thalassaemia[6] KLF1 is indeed also known to be involved in the expression of beta-globin[25], and the fact that the expression of KLF1 in-flight is somewhat reduced from the base expression observed pre-flight indicates that the production of adult beta-globins may be somewhat repressed, which would correlate with observations made in the expression of HBB, as aforementioned. Conversely, observations of expression of the MYB gene showed an overall reduction in expression in flight and a somewhat increased expression post-flight for the JAXA astronauts, with post-flight expression of MYB also increasing upon return for the Inspiration4 crew. Interestingly, a similar trend in expression is also seen in the NASA Twins study when considering the CD4 and CD8 cellular fractions, although an opposite observation was made in the flight expression of MYB measured for the LD cellular fraction. An in-flight repression and increased post-flight expression of MYB is in line with expectations for an in-flight adult to fetal globin switch, given that MYB is known to support erythropoiesis through activation of KLF1 and LMO2 expression[37], with KLF1 being a gamma-globin repressor.

EPO is one of the hallmark genes for the promotion of erythropoiesis, known to typically be up-regulated in hypoxic conditions[38]. It encodes for a glycosylated cytokine, erythropoietin, that binds to the erythropoietin receptor, promoting red blood cell production in the bone marrow[39]. EPO was found to be significantly down-regulated in-flight and up-regulated post-flight in the JAXA cohort, but remaining mostly at stable levels in both the Inspiration4 study and the NASA Twins study (for the latter, slight down-regulation post-flight was instead noted in the LD cell fraction). The observation made in the JAXA cohort seems to be in congruence with the observed phenotype of space anaemia, suggesting that the down-regulation of EPO in flight, as observed for the JAXA astronauts, might have had a negative effect on erythropoiesis and thus contributed to space anaemia. Observations of the EIF2AK1 gene show significant up-regulation in-flight in the JAXA astronauts, although similar observations were not made in the NASA Twins study. This gene is one of several known essential genes for erythropoiesis, and its activation under oxidative stress conditions has been shown to result in erythroid differentiation via activation of the ATF4 signalling pathway[40]. The ATF4 gene itself was found to be down-regulated post-flight, when compared to in-flight levels, for both the JAXA cohort and in the NASA Twins study, when considering the CPT cell fraction.

Other genes known to be heavily involved in erythropoietic pathways showed notable variation. Particularly, the ACVR1B gene showed significant down-regulation in-flight and up-regulation post-flight, both in the NASA Twins study (when looking at LD cell fractions) and in the JAXA cohort. With this gene known to be a positive regulator of differentiation of erythrocytes, these observations suggest that one possible manner in which overall red blood cell counts decrease in astronauts in microgravity could be through disruption of expression of such genes, resulting in slower rates of erythroid differentiation and hence maturity. Conversely, however, ARID1B was also shown to be

down-regulated in flight for the JAXA astronauts, with this gene known to be involved in erythropoiesis suppression. Indeed, knockdown on ARID1B has been shown to result in a hundred-fold increase in red blood cell count in hypoxic conditions[41]. ARID1B knockdown is also known to increase the expression of GATA1 by three-fold[41], with GATA1 being involved in the differentiation of embryonic stem cells to mature erythrocytes[42,43]. Thus, the down-regulation of ARID1B could be interpreted as a response mechanism for space anaemia, with its reduced expression possibly opposing the reduced red blood cell counts in astronauts in flight.

Observations of the prevalence of GATA1 expression showed some up-regulation in flight in the JAXA astronauts, with a more pronounced increase in GATA1 expression in this cohort seen post-flight. In the NASA Twins study, post-flight up-regulation was noted in GATA1 expression, when compared to pre-flight and in-flight levels in the LD cell fraction. This seems to suggest that if the increased expression of ARID1B had an ameliorating effect on space anaemia while in-flight in this case, this was likely achieved through pathways other than GATA1-mediated differentiation and erythroid maturation. On the other hand, the increased expression of ARID1B in flight might have also had a delayed response in increased GATA1 expression (observed post-flight) and thus amelioration of astronauts post-flight could be in part attributed to the observed GATA1 up-regulation. Conversely, the expression of LDB1 was observed to undergo up-regulation in-flight when compared with both pre-flight and post-flight levels in the NASA Twins study data, when considering the LD cellular fraction only. LDB1 is known to operate in synergy with other transcription factors such as KLF1, GATA1 and TAL1, facilitating the recruitment of LDB1-nucleated protein complexes to activation sites for alpha and beta-globin genes via long-distance interactions[44]. Since both alpha and beta-globin gene overall expression was actually observed to decrease in flight, particularly in the JAXA cohort, this may indicate that the up-regulated expression of LDB1 in flight may in fact be one method of counter-action for overall reduced adult globin genes observed in space anaemia.

It was indeed observed that while some transcription factors involved in erythropoiesis regulation showed in-flight variation in expression, others only showcased a change in expression post-flight. TAL1, a known haematopoiesis regulator and known to be essential for erythropoiesis[45-47], showed in-flight up-regulation and post-flight down-regulation in the JAXA cohort as well as in the NASA twin study (LD cell fraction). However, LMO2, which is also known to be involved in haematopoiesis and essential for erythropoiesis[47], showed increased expression post-flight only. The JAXA astronauts also showed a slight in-flight increase coupled with a post-flight down-regulation in the expression of NFE2, with a similar post-flight down-regulation observed in the NASA twin study (CD4 cell fraction). NFE2 plays a role in erythroid maturation, involved in controlling the transcription of erythroid-specific genes[48]. In addition, NFE2 is also involved in the activation of PBGD[49] and FECH[50], two enzymes involved in haem biosynthesis. This suggests that the up-regulation of NFE2 may be an in-flight response to counter space anaemia.

### The expression of miRNAs involved in erythropoiesis and globin expression

Micro RNAs (miRNAs) constitute an important regulatory mechanism at a post-transcriptional level, binding to messenger RNA (mRNA) and typically resulting in repression of expression, a mechanism termed gene silencing. A number of such miRNAs involved in the regulation of globin expression have been identified in previous studies. Apart from being involved in the regulation of globin gene expression, globin miRNAs have also been shown to play a role in the switch between embryonic and fetal haemoglobin, as well as the regulation of adult globin. Specifically, a number of miRNAs have been identified as having an effect on the expression of a number of transcription factors

involved in the induction of fetal haemoglobin. An example of such miRNAs includes miR-15a, miR-16-1, miR-23a, miR-26b, miR-27a and miR-451, which have been shown to modulate the effects of a number of transcription factors including BCL11A, MYB, KLF3 and Sp1, resulting in an increase in fetal haemoglobin production while suppressing beta-globin expression[51,52].

Globin miRNA levels measured from the NASA Twins study demonstrated significant changes in selected miRNA expression levels recorded during flight and post-flight. Figure fig:expression_mirnas_NASA shows normalised expression levels subdivided across a number of cellular fractions. Among these, the LD cellular fraction as well as the CPT fraction contain the erythroid portion of the cellular fractions here considered. A lower expression of nearly all miRNAs investigated was recorded during flight in these two fractions, except for miR-150-3p and miR-23a-5p, when compared against the expression recorded post-flight. Similar trends were also observed when assessing a number of miRNAs in the CD19 cellular fraction for expression before flight and expression after flight, while an opposite trend was observed for the CD4 cells, with higher pre-flight expression than post-flight expression. For the purposes of obtaining meaningful comparisons, ground twin expression values were not considered for discussion due to the high variation observed when comparing both twins pre-flight, suggesting strong individual variations.

The down-regulation observed in-flight, followed by a stronger expression post-flight in the erythroid cell fractions (LD and CPT) seems to suggest that the investigated miRNAs may indeed play a significant role in the observed globin gene expression values provided in Fig. fig:expression_mirnas_NASA. Indeed, the down-regulation of investigated miRNAs in the erythroid fractions coincides with the down-regulation of expression of adult globin genes on the alpha and beta-globin loci. Additionally, an increased expression of the same miRNAs on return to the ground after flight also coincides with an increased expression of alpha and beta-globin genes, as well as a number of transcription factors, a trend observed in the NASA Twins study, in the JAXA astronaut cohort and the Inspiration4 crew.

### Concluding remarks

It was observed that overall adult globin gene expression was repressed in flight, while fetal globin gene expression was enhanced. Several genes coding for transcription factors known to be involved in the fetal-to-adult haemoglobin switch also showed variable expression between pre-flight, in-flight and post-flight conditions. Indeed, such observations, discussed in this study, seem to indicate that an adult-to-fetal globin switch mechanism is indeed activated in-flight, with post-flight amelioration of space anaemia seeing a reversion of expression back to normal healthy adult levels.

Several of the observed changes followed an in-flight shift in favour of fetal haemoglobin, with increased expression of genes involved in gamma-globin expression and repression of genes normally involved in beta-globin gene expression, or involved in repression of gamma-globin genes themselves (such as BCL11A and KLF1). These observations seem to indicate that fetal haemoglobin expression is favoured in flight, as seen in several haemoglobinopathies where the expression of fetal haemoglobin tends to have an ameliorating effect. This suggests that the observed in-flight switch may be a similar mechanism, in this case, to combat the effects of space anaemia. In this regard, future studies will make direct measurements of fetal haemoglobin, to shed light on the direct effects of such known haemoglobin switch actors on the actual levels of fetal haemoglobin in spaceflight and afterward.

In addition, a number of genes that play a role in erythropoiesis were also found to be perturbed in flight, in some cases suggesting that the observed space anaemia condition may not only be a result of increased haemolysis but also due to down-regulation in the expression of certain genes known to play an important positive regulatory role in erythropoiesis. A number of other genes involved in erythropoiesis

were seen to increase in expression in-flight as well, possibly suggesting that these could act as a response mechanism, during flight itself, to the anaemic condition. Some of these genes, as with the globin switch mechanism, only showed a delayed expression variability in post-flight readings, suggesting that they might be involved in increasing erythropoiesis rates after landing. Indeed, astronauts have been observed to recover total erythroid cell counts in the weeks following return from spaceflight, effectively recovering from space anaemia.

While our study provides comprehensive insights into the impact of spaceflight on astronaut blood samples, we acknowledge the limitation regarding the diverse duration of exposure. Due to logistical constraints, our analysis primarily focuses on a specific time frame. Future research with extended mission duration would undoubtedly enhance our understanding of the temporal dynamics involved, and we encourage further exploration in this area to broaden the scope of our findings.

## Methods

### NASA twins study RNA-seq analysis

Specific details related to all the methods for the NASA Twins Study RNA-seq data can be found in the following references[9,53]. Briefly, the NASA Twins Study involved two male twin subjects, aged 50 years old at the time of launch. The flight twin spent 340 days aboard the ISS while his identical twin stayed on Earth as the ground control.

Blood samples were collected into 4 mL CPT tubes (BD Biosciences Cat No.362760) as per the manufacturer's recommendations. Cell separation was performed by centrifugation at $1800 \times g$ for 20 min at room temperature, both on the ISS and for the ground-based samples. Ambient blood collected samples slated for immediate return on Soyuz capsule were stored at 4 °C until processing (average of 35–37 h after collection, including repatriation time). Samples collected on Earth and the ISS and planned for long-term storage were mixed by inversion and immediately frozen at −80 °C.

Fresh processing of CPT tubes was performed as follows. Firstly, plasma was retrieved from the CPT tubes and flash frozen prior to long-term storage at −80 °C. Secondly, the peripheral blood mononuclear cells (PBMCs) were recovered and washed in PBS. 0.5 million PBMCs were retrieved from one pre-flight and one post-flight sample, pelleted, flash frozen and stored at −80 °C until use for RNA extractions. The flow through from cell sorting steps was recovered as the lymphocyte depleted (LD) fraction; LD and PBMC cell specimens were lysed into RLT+ buffer (Qiagen Cat No.1053393), flash frozen and stored at −80 °C until use.

Cells were isolated and prepared using the BD Rhapsody platform according to the BD Rhapsody Express Single-Cell Analysis System Instrument User Guide, with a custom-designed RNA and epitope panel. Briefly, cells from each sample were labelled with sample tags and then pooled after being washed twice with FACS buffer. Combined samples were then washed an additional time before being stained with the BD AbSeq Ab-Oligo reagents. After staining, the cells were washed twice before resuspension at approximately 20,000 cells in 620 $\mu$L. These cells were then isolated using the Single-Cell Analysis System and cDNA Synthesis with the BD Rhapsody Express Single-Cell Analysis System using the manufacturers' protocol (BD Biosciences). Cells were loaded onto 3 BD Rhapsody nanowell cartridges. Cartridges were loaded with Cell Capture Beads (BD Biosciences) before shaking for 15 s at 1000 rpm. Cells were lysed and cell capture beads were retrieved and washed prior to Exonuclease I treatment and reverse transcription.

Targeted amplification of cDNA with the Human Immune Response Panel primers and custom supplemental panel was done through 10 PCR cycles. PCR products were purified, and mRNA and AbSeq products were separated by SPRIselect beads with double-sided selection. mRNA products were amplified further with 15 PCR cycles. Final libraries were indexed with 8 PCR cycles. Library quality was

assessed by Bioanalyzer (D1000 HS, Agilent). Library DNA concentration was quantified using a Qubit dsDNA HS Kit (ThermoFisher, No.Q32854) on a Qubit Fluorometer. Libraries were diluted to 2 nM and multiplexed before being sequenced on three lanes of a Novaseq-6000. The mean read depth per cell was 13,244.31 mRNA reads per cell and 11,506.98 AbSeq reads per cell for a combined 243,751.30 reads per cell.

FASTQ files were uploaded to Seven Bridges Genomics. Data was demultiplexed and sequences analysed with BD's Rhapsody pipeline (BD Rhapsody Analysis Pipeline 1.4 Beta) on Seven Bridges (www.sevenbridges.com). Data was then loaded into Seurat (version 3.2.0) for analysis[54]. This generated a sparse matrix file of features by barcodes. This sparse matrix data was then read into R using the R package Seurat 3.2.0, and standard quality control was run to remove cells with few genes. Data was then scaled and normalised. MAST[55] was used to determine DEGs between all combinations of pre-flight, return and post-flight for both TW and HR samples. Heatmaps were made for the curated globin genes on the DEG values per time point using the R package pheatmap version 1.0.12.

### NASA twins study miRNA-Seq analysis
Specific details related to all the methods related to the NASA Twins Study miRNA-seq data can also be found in the provided references[9,53]. We will briefly highlight the key methods. Small RNA libraries were prepared from 50ng total RNA using the NEBNext Multiplex Small RNA Library Prep Set for Illumina (NEB No.E7560) per manufacturer's recommendations with the following modifications: adaptors and RT primer were diluted fourfold, 17 cycles of PCR were used for library amplification, and no size selection was performed. The i7 primers in the NEBNext Multiplex Oligos for Illumina Dual Index Primers (NEB No.E7600, NEB No.E7780) were used to supplement the index primers in NEB No.E7560. The libraries were sequenced in an Illumina NextSeq instrument (1 × 50bp).

Standard libraries were pre-processed and quality-controlled using miRTrace[56]. Subsequently, reads were mapped against MirGen-eDB sequences[57] using the miRDeep2[58] quantifier module. Expression values were normalised to reads per million (RPM) considering only miRNA counts. The normalised RPM value was utilised for all analysis. If the value for the miRNA was zero for all samples that miRNA was excluded from the analysis. To determine statistically significant miRNAs, ANOVA analysis with $p$-value < 0.2 was independently performed for each cell type.

For flight-only comparisons, the same statistical significance was applied for all time points excluding the ground samples and all ambient return samples. All statistics were run independently for each cell condition/type. To determine the overlap of the miRNA signature in this paper with the Twins Study miRNA-seq data, the proposed miRNAs were used as well as all mature components of the miRNAs included in the miRNA family. Specific miRNAs related to globin pathways were plotted as heatmaps on the expression values using the R package pheatmap version 1.0.12.

### JAXA CFE epigenome study: RNA quantification data
Aggregated RNA differential expression data and study protocols were shared through NASA's Open Science Data Repository with accession number: OSD-530[59]. Plasma cell-free RNA samples for RNA-seq analysis were derived from blood samples collected from 6 astronauts before, during, and after the spaceflight on the ISS. Mean expression values were obtained from normalised read counts of 6 astronauts for each time point. Heatmaps were computed for the curated globin genes on the normalised values per time point using the R package pheatmap version 1.0.12. Differential expression analysis was carried out with DESEQ2, and subsequent volcano plots were plotted using Python's Matplotlib package. Principal component analysis was carried out using Python's sklearn package.

### Inspiration4: data collection and analysis
Detailed methods for sample collection and data processing are described in ref. 60. In summary, blood samples were collected before (pre-launch: L-92, L-44, and L-3) and after (return; R+1, R+45, and R+82) the spaceflight. Chromium Next GEM Single Cell 5' v2, 10x Genomics was used to generate single-cell data from isolated PBMCs. Sub-populations were annotated based on Azimuth human PBMC reference. The Seurat R package was used to normalise RNA count data and calculate the average expression of each gene.

For the plasma cf-RNA, plasma samples were collected in cf-DNA Blood Collection (Streck: #230470) tubes at each time point and stored at −80 °C. Plasma samples were thawed at room temperature and subsequently centrifuged at 1300 × $g$ for 10 min at 4 °C. cfRNA was isolated from the plasma supernatant (300–800 μL) using the Norgen Plasma/Serum Circulating and Exosomal RNA Purification Mini Kit (Catalogue No. 51000, Norgen). Next, 10 mL of DNase Turbo Buffer (Catalogue No. AM2238, Invitrogen), 3 mL of DNase Turbo (Catalogue No. AM2238, Invitrogen), and 1 mL of Baseline Zero DNase (Catalogue No. DB0715K, Lucigen-Epicentre) was added to the extracted RNA and incubated for 30 min at 37 °C. Subsequently, the treated RNA was concentrated into a final volume of 12 mL with the Zymo RNA Clean and Concentrate Kit (Catalogue No. R1015, Zymo).

Sequencing libraries were prepared from 8 μL of concentrated RNA using the Takara SMARTer Stranded Total RNA-Seq Kit v3−Pico Input Mammalian (634485, Takara) and barcoded using the SMARTer RNA Unique Dual Index Kit (634451, Takara). Library concentration was quantified using a Qubit 3.0 Fluorometer (Q33216, Invitrogen) with the dsDNA HS Assay Kit (Q32854, Invitrogen). Libraries were quality-controlled using an Agilent Fragment Analyser 5200 (M5310AA, Agilent) with the HS NGS Fragment kit (DNF 474-0500, Agilent). Libraries were pooled to equal concentrations and sequenced at the Cornell Genomics Core on an Illumina NextSeq 2000 machine using 150-base pair, paired-end sequencing for an average of 26 million reads per sample.

All human astronaut subjects have consented at an informed consent briefing (ICB) at SpaceX (Hawthorne, CA), and samples were collected and processed under the approval of the Institutional Review Board (IRB) at Weill Cornell Medicine, under Protocol 21-05023569. All crew members have consented to data and sample sharing. JAXA and NASA astronaut data dissemination follows agency protocols.

### Reporting summary
Further information on research design is available in the Nature Portfolio Reporting Summary linked to this article.

## Data availability
The JAXA CFE data is available via the NASA Open Science Data Repository's (OSDR)'s Biological Data Management Environment (https://osdr.nasa.gov/bio/) with accession numbers: OSD-5302, DOI: 10.26030/r2xr-h714. Deposited data from the sequencing data from the NASA Twin Study can be found on the NASA Life Sciences Data Archive (LSDA) and the accession code is not available due to privacy concerns. LSDA is the repository for all human and animal research data, including that associated with this study. LSDA has a public-facing portal where data requests can be initiated (https://nlsp.nasa.gov/explore/lsdahome/datarequest). The LSDA team provides the appropriate processes, tools, and secure infrastructure for the archival of experimental data and dissemination while complying with applicable rules, regulations, policies, and procedures governing the management and archival of sensitive data and information. The LSDA team enables data and information dissemination to the public or to authorised personnel either by providing public access to information or via an approved request process for information and data from the LSDA in accordance with NASA Human Research Programme and JSC Institutional Review Board direction. The Inspiration4 data has been uploaded to two data repositories: the NASA Open Science Data

Repository (osdr.nasa.gov; comprised of NASA GeneLab and the NASA Ames Life Sciences Data Archive (ALSDA)), and the TrialX database. Identifiers for publicly downloadable datasets in the OSDR are documented as follows: Data can be visualised online through the SOMA Browser (https://epigenetics.weill.cornell.edu/apps/l4_Multiome/), the single-cell browser (https://soma.weill.cornell.edu/apps/l4_Multiome/), and the microbiome browser (https://soma.weill.cornell.edu/apps/l4_Microbiome/). For the PBMC data, the data is available with OSDR accession ID: OSD-570 and the following link: https://osdr.nasa.gov/bio/repo/data/studies/OSD-570/. All source data used to produce the figures in this study are provided in the Source Data file. Source data are provided in this paper.

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

## Acknowledgements

We would like to extend our gratitude to the Japan Aerospace Exploration Agency (JAXA) for the data on their Japanese astronauts. We also thank the SpaceX Inspiration4 astronauts for their contribution of blood samples and data which were essential for our analysis. We are grateful to Dr. Wasim Ahmed, CEO of Metavisionaries, for the scientific funding support provided to Dr. Josef Borg, facilitated by the Research, Innovation and Development Trust (RIDT) foundation at the University of Malta, Malta. CEM also thanks the Scientific Computing Unit (SCU), the WorldQuant and GI Research Foundation, NASA (NNX14AH50G, NNX17AB26G, NNH18ZTT001N-FG2, 80NSSC22K0254, 80NSSC23K0832), the National Institutes of Health (R01ES032638, U54AG089334), and the LLS (MCL7001-18, LLS 9238-16, 7029-23), and Boryung for their financial support and research guidance as well.

## Author contributions

Dr. Josef Borg, Mr. Conor Loy, and Dr. JangKeun Kim performed the experiments; contributed reagents, materials, analysis tools, or data; and wrote the paper. Dr. Christopher Mason, Dr. Afshin Beheshti, and Prof. Joseph Borg conceived and designed the experiments; performed the experiments; analysed and interpreted the data; contributed reagents, materials, analysis tools, or data; and wrote the paper. Dr. Alfred Buhagiar, Mr. Christopher Chin, Ms. Namita Damle, Prof. Iwijn De Vlaminck, Prof. Alex Felice, Ms. Tammy Liu, Ms. Irina Matei, Dr. Cem Meydan, Dr. Masafumi Muratani, Mr. Omary Mzava, Dr. Eliah Overbey, Ms. Krista Ryon, Dr. Scott Smith, Dr. Braden Tierney, Dr. Guy Trudel, and Dr. Sara Zwart interpreted the data; contributed reagents, materials, analysis tools, or data; and wrote the paper.

## Competing interests

The authors declare no competing interests.
