## [Peer Review File · Nature Communications]

Spatiotemporal expression and control of haemoglobin in spaceReviewers' comments:

Reviewer #1 (Remarks to the Author):

This is a descriptive study of potentially very interesting data concerning the effects of microgravity experienced during stay at the international space station on erythroid parameters. It can be accepted that the data set used is incomplete and imperfect. For instance, gene expression could not be measured in a pure population of erythroid progenitors and many of the blood parameters could not be determined in space (Figure 5). In this context, I wonder whether the single cell analyses (BD Rhapsody Express Single-Cell Analysis System) might provide a more in-depth reading of changes occurring in the erythroid compartment during space flight. For the data in Figure 5, I wonder why levels of erythropoietin were not included. These can be easily measured in serum using a commercially available ELISA assay. Were reticulocyte levels measured? These are a sensitive indicator of compensated anemia. Currently, it is not clear whether the decreased hematocrit and RBC count are a physiological adaptation to conditions in space; I assume much less energy (and hence reduced oxygen consumption) is required to go about daily life in space. I see no evidence for increased hemoanalysis. If hemoglobin switching has a role, is this reflected in HbF levels? The fold-changes presented are qualitative measurements, it is not clear whether these translate into physiologically relevant quantitative changes in Hb composition. Finally, I wonder whether a more holistic approach such as PCA or tSNE analysis would be helpful in visualizing the individual pre-, in-flight and post-flight samples. Are they separated according to flight status?

Some other questions:

Are the subjects of similar age and ethnicity? Are all subjects male?

Statistical underpinning seems to be weak, e.g. significant down-regulation or up-regulation (p-value < 0.1). This would normally be considered an acceptable threshold for significance.

As far as I know, nobody has ever been able to show theta-globin protein. This alpha-like globin has no known physiological function.

The Discussion section is far too long (page 10-18), in sharp contrast to the Results section (page 6-9). The Results section should be rewritten explaining the rationale, experimental approach, main outcomes and conclusions of the experiments performed.

Reviewer #2 (Remarks to the Author):

Borg et al, presents an interesting analysis of RNAseq data obtained from multiple space missions. They used the RNAseq data to pick out key regulatory genes that are involved in erythropoiesis and hemoglobin expression/switching at different time points of the astronaut in space. The authors did a comprehensive analysis of the RNAseq data from the NASA space mission in the hematopoiesis context. The authors went ahead and described the analysis which explains consequences of the anemia in the astronauts in zero-gravity. I think this article states what has been already demonstrated in previous analysis on the loss of anemia but expounds on the direct consequences. But a deeper insight on what could be the cause of the anemia in the environment would be ideal. In addition most of the consequences they mentioned seemed to be driven by the obvious of stress erythropoiesis driven by the anemia, which accounts for the globin switching and erythropoiesis pattern changes, which the authors did not mention. Below are some thoughts that could help strengthen the manuscript.

Revisions:

1. One hypothesis presented for the anemia in the astronauts, is due to the lack of gravity, but are there any other potential causes for the induced anemia, ie could be the increase exposure to radiation account for the oxidative stress leading to anemia. The authors could possibly look into any other cellular responses caused by other factors that might cause the anemia seen in astronauts?

2. Through the author's analysis, they implicated quite a few genes that are involved in globin switching, which is a hallmark sign of potential stress erythropoiesis. Does the authors think that the outcomes of the response to the anemia are direct responses of stress erythropoiesis, in which

could they use this data to further elucidate aspects of the phenomenon.

3. They saw an increase of EIF2AK1, which is also known as HRI, an important gene implicated in stress erythropoiesis.

Minor revisions:

1. In line 327, the authors referred to figure x, please correct.

To the reviewers,

We would like to thank you for your comments concerning our manuscript. In this revision, we provide further analyses and figures concerning results obtained from our various datasets and tackled certain aspects previously highlighted. The new version should provide a better grounding for the conclusions drawn while providing more focus on the obtained results.

More specifically, our revisions focus on the following:

(a) We have included principal component analysis to showcase separation of cases by flight status in the JAXA astronaut cohort as well as the Inspiration 4 crew. In both cases, separation by flight status is evident, with separate clustering of pre-flight, in-flight and post-flight samples. In particular, this was carried out using genes and trans-acting factors of interest (globin genes and trans-acting factors known to be involved in erythropoiesis, hematopoiesis and the globin gene switch mechanism) as the features for the PCA.

(b) We provide a better statistical significance grounding for the analyzed datasets, specifically by providing breakdown of differential expression p-values for the NASA twin study – with those genes showing a p-value lower than 0.05 separately indicated. In addition, we provide volcano plots showcasing significantly differentially expressed genes ($p\text{-val} < 0.05$) with a \log_2 fold change threshold > 1.5 or < -1.5 (for up-regulated and down-regulated genes sets respectively).

(c) We have made further reference to previous studies concerning prevalence of space anaemia, as well as the observation of specific parameters.

In addition, for the sake of clarity of reporting the obtained results, we have also included a summary diagram detailing an overview of the observed trends in this study. This figure (included in the new manuscript version as Figure 1) is summarized as follows in the following excerpt from the new manuscript: *Notably, repressors for fetal haemoglobin (HBF) in the haemoglobin gene switch showed in-flight downregulation in almost all cases for both the NASA twins study and the JAXA astronaut cohort, suggesting an in-flight shift to favour production of HBF. Erythropoiesis promoters showed a mixed in-flight and post-flight expression regulation, with some promoters down-regulated in flight and thus likely contributing to the space anaemia phenotype while other promoters were up-regulated in flight, possibly in response to space anaemia. Erythropoiesis repressors were down-regulated in flight, possibly as a response to the space anaemia condition as well.*

We also include hereunder a response to the reviewers' comments, to ensure clarity of changes and updates that have now been included in the manuscript. We would also like to take this opportunity to thank the reviewers for their time and constructive comments, which we have taken on board and included updates for, wherever possible. We note also that some of the suggested work concerning limitations of this study will be the subject of follow-up work involving future space missions and studies which our group will be presently

embarking upon, particularly in cases where specific datasets were unfortunately not at our disposal at this time since particular measurements have not as yet been made (most importantly in flight).

We thank you once more for your consideration and time, and we look forward to your response.

Josef Borg

Reviewers' Comments:

Reviewer 1:

This is a descriptive study of potentially very interesting data concerning the effects of microgravity experienced during stay at the international space station on erythroid parameters. It can be accepted that the data set used is incomplete and imperfect. For instance, gene expression could not be measured in a pure population of erythroid progenitors and many of the blood parameters could not be determined in space (Figure 5). In this context, I wonder whether the single cell analyses (BD Rhapsody Express Single-Cell Analysis System) might provide a more in-depth reading of changes occurring in the erythroid compartment during space flight.

We thank the reviewer for their comment concerning the interest of the data and for acknowledging the potential significance of our study on the effects of microgravity on erythroid parameters during spaceflight. It is indeed the case that we had to work with the data available across three separate missions that included NASA's twin study, SpaceX Inspiration4 and JAXA, which meant that certain key parameters of interest were not, for example, measured in flight consistently for all. We do however want to bring to the attention of the reviewer that, as illustrated in Figure 1 of the paper, we undertook efforts to enhance the resolution of our study by separating cell fractions. Specifically, we utilized lymphocyte-depleted (LD) and separate lymphocyte fractions, in addition to an unsorted (CPT) fraction. This approach, depicted in our experimental design, aimed to provide a more detailed view than conventional bulk RNAseq. We believe that the fractionation strategy, as detailed in Figure 1, does contribute valuable granularity to our analysis. Ultimately, our aim is to attempt to obtain more and better data in future planned missions, in order to supplement the data showcased in this study.

For the data in Figure 5, I wonder why levels of erythropoietin were not included. These can be easily measured in serum using a commercially available ELISA assay. Were reticulocyte levels measured? These are a sensitive indicator of compensated anaemia. Currently, it is not clear whether the decreased hematocrit and RBC count are a physiological adaptation to

conditions in space; I assume much less energy (and hence reduced oxygen consumption) is required to go about daily life in space. I see no evidence for increased haemolysis.

Indeed, both erythropoietin and reticulocyte levels were included in a separate study: Trudel, Guy, et al. "haemolysis contributes to anaemia during long-duration space flight." *Nature Medicine* 28.1 (2022): 59-62. The advent of haemolysis is indeed well documented in this paper as well, and thus this study is building on the observations indicated in the referenced manuscript.

If haemoglobin switching has a role, is this reflected in HbF levels? The fold-changes presented are qualitative measurements, it is not clear whether these translate into physiologically relevant quantitative changes in Hb composition.

While we did perform quantitative measurements of mRNA levels for all globins using RNAseq, we acknowledge the importance of complementing these results with quantitative assessments of HbF levels. We recognize that our current presentation primarily focused on qualitative fold-changes, and we agree that translating these changes into physiologically relevant quantitative alterations is pivotal for a comprehensive understanding. In response to your query, we would like to assure the reviewer that steps are actively being taken to address this aspect in subsequent space missions. Specifically, our future research endeavors include dedicated measurements to provide quantitative insights into fetal haemoglobin levels by HPLC. Our current study has now shown that such measurements (as proposed by the reviewer) are indeed warranted and should be proposed since they will contribute an unprecedented set of data to the understanding of haemoglobin composition under microgravity conditions.

Finally, I wonder whether a more holistic approach such as PCA or tSNE analysis would be helpful in visualizing the individual pre-, in-flight and post-flight samples. Are they separated according to flight status?

We concur with the reviewer that such a holistic approach in visualizing whether there is separation by flight status is indeed of importance. This has now been included in the new version of the paper, with principal component analysis from both the JAXA and Inspiration4 astronaut cohorts. Indeed, there is separation by flight status for the different astronauts, as showcased by the obtained PCA plots. Notably, we used only genes of interest – those known to be involved in globin production, erythropoiesis, hematopoiesis and the adult to fetal globin switch mechanism – in order to eliminate the possibility that such clustering occurred due to other unrelated genes.

Are the subjects of similar age and ethnicity? Are all subjects male?

We would like to assure the reviewer that the subjects in our study encompass a diverse demographic, including individuals of varying ages, ethnicities, and genders. Specific details can be found in the methods section of our paper. Furthermore, we would like to highlight that the names of the astronauts involved, such as Mark and Scott Kelly (NASA astronauts twins study) and the SpaceX Inspiration4 crew, are in the public domain. This information contributes to the transparency of our study and allows for further scrutiny of the demographic aspects.

Statistical underpinning seems to be weak, e.g. significant down-regulation or up-regulation (p-value < 0.1). This would normally considered an acceptable threshold for significance.

Indeed, we appreciate that the normal accepted p-value threshold is 0.05, and have updated the NASA twin study diagram accordingly to showcase, specifically, which genes are significantly up-regulated or down-regulated in which cellular fraction. We still also include the 0.1 p-value as these genes could still merit further study and discussion, but we recognize that focus should be directed to the more significantly differentially expressed genes.

As far as I know, nobody has ever been able to show theta-globin protein. This alpha-like globin has no known physiological function.

This was indeed a mistake in the original manuscript, and has thus been rectified accordingly.

The Discussion section is far too long (page 10-18), in sharp contrast to the Results section (page 6-9). The Results section should be rewritten explaining the rationale, experimental approach, main outcomes and conclusions of the experiments performed.

We have shortened the discussion section and re-written and included further text in the Results section accordingly. Some text previously included in the results section has now also been moved to the results section to further clarify the rationale and outcomes of the obtained results. As indicated above, further plots have been added to results section (specifically, the addition of volcano plots for the JAXA cohort, a heatmap showcasing z-scores for normalized expression across genes of interest for the Inspiration4 crew and PCA plots for both the JAXA and Inspiration4 astronauts) while changes have been made to the NASA twin study plot to better showcase significantly expressed genes in different cellular fractions.

Reviewer 2:

Borg et al, presents a in interesting analysis of RNAseq data obtained from multiple space missions. They used the RNAseq data to pick out key regulatory genes that are involved in erythropoiesis and haemoglobin expression/switching at different time points of the astronaut in space. The authors did a comprehensive analysis of the RNAseq data from the NASA space mission in the hematopoiesis context. The authors went ahead and described the analysis which explains consequences of the anaemia in the astronauts in zero-gravity. I think this article states what has been already demonstrated in previous analysis on the loss of anaemia but expounds on the direct consequences. But a deeper insight on what could be the cause of the anaemia in the environment would be ideal. In addition most of the consequences they mentioned seemed to be driven by the obvious of stress erythropoiesis driven by the anaemia, which accounts for the globin switching and erythropoiesis pattern changes, which the authors did not mention.

We thank the reviewer for their comment concerning the interest of this study, which utilizes data from multiple missions. Indeed, our manuscript builds on previous studies making observations of anaemia and haemolysis in different astronauts. While the observations made in our study seem to implicate stress erythropoiesis leading directly to globin switching changes in erythropoiesis-related genes, this study constitutes a first attempt at showing a globin switch mechanism being activated in the space environment on board the ISS due to the stressors that result in stress erythropoiesis in the first place. RNAseq data analysis in this context has not previously been carried out using astronaut data, which is indeed one of the main aims of this study in and of itself, and thus this study aims to provide a direct observation of up-regulation and down-regulation of specific genes of interest in this context.

Below are some thoughts that could help strengthen to manuscript.
Revisions:

1. One hypothesis presented for the anaemia in the astronauts, is due to the lack of gravity, but are there any other potential causes for the induced anaemia, ie could be the increase exposure to radiation account for the oxidative stress leading to anaemia. The authors could possibly look into any other cellular responses caused by other factors that might cause the anaemia seen in astronauts?

Indeed, the increased exposure to radiation on board the ISS might play some role in oxidative stress leading to anaemia. Further investigation into the specific effects of radiation on erythroid parameters and haemolysis could provide a more complete understanding. However, there are also other factors, which are beyond the immediate scope of this publication, that too may act on space anaemia and erythropoiesis. These include; (1) Microgravity-Induced Changes in Bone Marrow. Altered bone marrow function due to the lack of bone stimulation in space is a known cause. Investigating the impact of microgravity on

bone marrow composition, adipose tissue accumulation, and erythropoiesis could shed light on additional mechanisms of space anaemia. (2) Mitochondrial Stress and Dysregulation; There is a possibility of shortened mature RBC lifespan due to mitochondrial stress. Looking closer and more in depth at the mitochondrial function in red blood cells could be a crucial avenue for further research. (3) Genetic and Epigenetic Factors; which we have strongly alluded too in our research. It is of course possible to delve even deeper into the genetic and epigenetic factors influencing erythropoiesis in space which could provide insights into individual variations in astronauts' responses. (4) Environmental Factors; Besides microgravity and radiation, other environmental factors in space, such as altered atmospheric conditions, could influence erythroid parameters. Investigating the comprehensive space environment and its impact on hematological parameters is essential. (5) Intravascular vs. Extravascular haemolysis; further clarification on the specific sites and mechanisms of haemolysis in space is crucial for developing targeted mitigation strategies. And finally (6) the effects of Long-Term Space Exposure; one can only hypothesize that the longer humans stay in space, the more severe the anemic phenotype will be.

2. Through the author's analysis, they implicated quite a few genes that are involved in globin switching, which is a hallmark sign of potential stress erythropoiesis. Does the authors think that the outcomes of the response to the anaemia are direct responses of stress erythropoiesis, in which could they use this data to further elucidate aspects of the phenomenon.

We have indeed found several genes known to be typically implicated in stress erythropoiesis that showcase varying levels of dysregulation in spaceflight, together with recovery being instigated post-flight.

3. They saw an increase of EIF2AK1, which is also known as HRI, an important gene implicated in stress erythropoiesis.

Indeed, this observation has been made and is now referenced in text as follows: *Observations of the EIF2AK1 gene show significant up-regulation in-flight in the JAXA astronauts, although such similar observations were not made in the NASA Twins study. This gene is one of several known essential genes for erythropoiesis, and its activation under oxidative stress conditions has been shown to result in erythroid differentiation via activation of the ATF4 signalling pathway.*

Minor revisions:

1. In line 327, the authors referred to figure x, please correct.

This has been corrected accordingly.

REVIEWERS' COMMENTS

Reviewer #1 (Remarks to the Author):

The authors have addressed my questions satisfactorily. The paper has been significantly improved after revision.